# Constrained deep neural network architecture search for IoT devices accounting for hardware calibration

**Florian Scheidegger**[1,2]     Luca Benini[1,3]     Costas Bekas[2]     Cristiano Malossi[2]

[1] ETH Zürich, Rämistrasse 101, 8092 Zürich, Switzerland

[2] IBM Research - Zürich, Säumerstrasse 4, 8803 Rüschlikon, Switzerland

[3] Università di Bologna, Via Zamboni 33, 40126 Bologna, Italy

## Abstract

Deep neural networks achieve outstanding results for challenging image classification tasks. However, the design of network topologies is a complex task, and the research community is conducting ongoing efforts to discover top-accuracy topologies, either manually or by employing expensive architecture searches. We propose a unique narrow-space architecture search that focuses on delivering low-cost and rapidly executing networks that respect strict memory and time requirements typical of Internet-of-Things (IoT) near-sensor computing platforms. Our approach provides solutions with classification latencies below 10 ms running on a low-cost device with 1 GB RAM and a peak performance of 5.6 GFLOPS. The narrow-space search of floating-point models improves the accuracy on CIFAR10 of an established IoT model from 70.64% to 74.87% within the same memory constraints. We further improve the accuracy to 82.07% by including 16-bit half types and obtain the highest accuracy of 83.45% by extending the search with model-optimized IEEE 754 reduced types. To the best of our knowledge, this is the first empirical demonstration of more than 3000 trained models that run with reduced precision and push the Pareto optimal front by a wide margin. Within a given memory constraint, accuracy is improved by more than 7% points for half and more than 1% points for the best individual model format.

## 1 Introduction

Designing an economically viable artificial intelligence system has become a formidable challenge in view of the increasing number of published methods, data, models, newly available deep-learning frameworks as well as the hype surrounding special-purpose hardware accelerators as they become commercially available. The availability of large-scale datasets with known ground truths [12, 42, 13, 51, 28, 10, 33, 54, 9, 5, 34, 37] and the widespread commercial availability of higher computational performance—usually achieved with graphic-processing units (GPUs)—has driven the current growth of and strong interest in deep learning and the emergence of related new businesses. Smart homes [29], smart grids [15] and smart cities [17] trigger a natural demand for the Internet of Things (IoT), which are products designed to be low in cost and feature low energy consumption and fast reaction times due to the inherent constraints given by final applications that typically demand autonomy with long battery lifetimes or fast real-time operation. Experts estimate that there will be some 30 billion IoT devices in use by 2020 [35], many of which serve applications that benefit from artificial intelligence deployment.

In this context, we propose an automatic way to design deep-learning models that satisfy user-defined constraints specifically tailored to match typical IoT requirements, such as inference latency bounds. Additionally, our approach is designed in a modular manner that allows future adaptations and

specialization for novel network topology extensions to different IoT devices and lower precision contexts. Our main contributions are the following:

- We propose an end-to-end approach to synthesize models that satisfy IoT application and hardware constraints.
- We propose a narrow-space architecture search algorithm to leverage knowledge from large reference models to generate a family of small and efficient models.
- We evaluate reduced precision formats for more than 3000 models.
- We isolate IoT device characteristics and demonstrate how our concepts operate with analytical network properties and map them to final platform-specific metrics.

The remainder of this paper is organized as follows. Section 2 describes related work, Section 3 introduces the core design procedures, Section 4 details and merges a full synthesis workflow, Section 5 presents and discusses the obtained results, and Section 6 concludes our findings.

## 2 Related work

Automated architecture search has the potential to discover better models [31, 52, 53, 55, 56, 6, 4, 49]. However, traditional approaches require a vast amount of computing resources or cause excessive execution times due to the full training of candidate networks[38]. Early stopping based on learning-curve predictors [14] or transferring learned wights shortens run-times [48]. A method called Train-less Accuracy Predictor for Architecture Search (TAPAS) demonstrates how to generalize architecture search results to new data without having to train during the search process [26]. Architecture searches face the common challenge of defining the search space. Historically, new networks were developed independently by expert knowledge that outperforms previously found networks generated by architectural searches. In such cases, very expensive reconsiderations led to follow-up work to account correctly for a richer search space [36, 47]. Recent progress in the field, such as MnasNet [46] and FBNet [50], tailor the search by optimizing a multi-objective function including inference time on smartphones. MnasNet trains a controller that adjusts to more optimal sample models in terms of multi-objectivity. FBNet trains a supernet by a differentiable neural architecture search (DNAS) in a single step and claims to be $420\times$ faster by avoiding additional model training steps. In contrast to solving a joint optimization problem in one step, our proposed union of narrow-space searches takes a modular approach that separates the search process of finding architectures that strictly satisfy constraints from the training of candidate networks. That way, we can analyze 10,000 architectures with no training cost and select only a small subset of suitable candidates for training.

Compression, quantization and pruning techniques reduce heavy computational needs based on the inherent error resilience of deep neural networks [39]. Mobile nets [22] or low-rank expansions [27] change the topology into layers that require fewer weights and reduce workloads. Quantization studies the effect of using reduced precision floating-point or fixed-point formats [21, 30], whereas compression attempts to reduce the binary footprint of activation and weight maps [7]. Pruning approaches avoid computation by enforcing sparsity [3]. We use floatx, an IEEE 754-compliant reduced precision library [16], to assess data format-specific aspects of networks. The novelty of our work is that we jointly evaluate network topologies in combination with reduced precision.

## 3 Core design procedures

### 3.1 Architecture search

It is challenging to define a space $S$ that produces enough variation and simultaneously reduces the probability of sampling suboptimal networks. We propose narrow-space architecture searches, where results are obtained by aggregating $n$ independent searches $S = \bigcup_{i=1}^{n} S_i$. As a good search space should satisfy $S_r \subset S$, where $S_r = \{M_1, ..., M_n\}$ is a set of reference models, we construct $S$ by designing narrow spaces that obey $M_i \in S_i$ in order to guarantee $S_r \subset S$. Instead of considering one large space, we have specialized search spaces that produce simple sequence structures with residual bypass operations (ResNets [19]) to even high fan-out and convergent structures such as they occur in the Inception module [44] or DenseNets [24]. Aggregation allows results to be extended easily with a tailored narrow-space search for new reference architectures. Next, we define a set

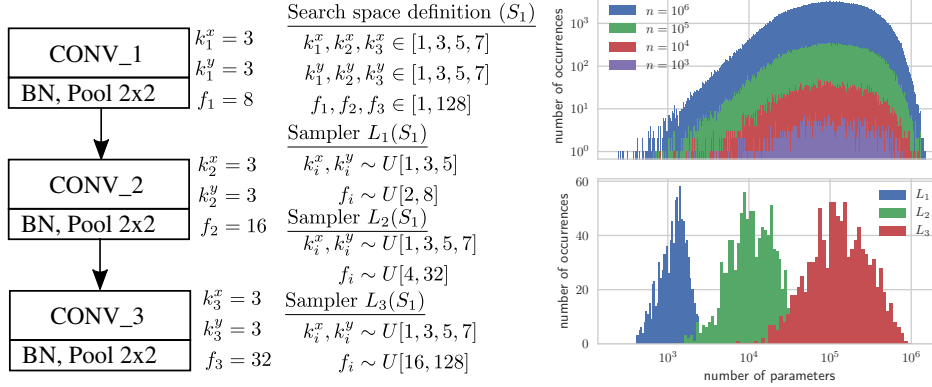

Figure 1: Left: Three-layer architecture. Middle: Default configuration of search space with restricted sampling laws. Right: Statistics of number of parameters obtained by sampling up to one million networks from the base configuration space and 1000 networks from the restricted sampling laws.

of distribution law configurations $L_1(S_i), ..., L_k(S_i)$ that allow samples to be drawn in a biased way such that models satisfy the properties of interest. Figure 1 illustrates the advantages over a uniform distribution among valid networks. Consider a space of three-layer networks with allowed variations in kernel shapes in $\{1, 3, 5, 7\}$ and output channels in $[1, 128]$ leading to $|S| = 4^6 * 128^3 = 8.6 * 10^9$ network configurations.

Figure 1 shows the statistics for up to $10^6$ samples compared with sampling only 1000 samples using restricted samplers $L_1, L_2$ and $L_3$. Restricted random laws efficiently generate networks of interest, in contrast to a uniform sampler that fails to deliver high sampling densities in certain regions. For example, only 132 out of $10^6$ networks have fewer than 1000 parameters.

We define each narrow-space architecture search and its sampling laws according to the following design goals: First, only valid models are generated with a topology that resembles and includes the original model. Second, the main model-specific parameters are varied, and efficient models are obtained mainly by lowering channel widths in convolutional layers and reducing the number of topological replications. Third, all random laws are defined following a uniform distribution over available options, where the lower and upper limits were used as a way to bias the models to span several orders of magnitude targeting the range of parameters and flop counts relevant for IoT applications.

## 3.2   Precision analysis

Precision analysis evaluates model accuracies for models having reduced precision representations. Following general methodology, we perform precision analyses on the backend device that has different execution capabilities than current or future targeted IoT devices. This methodology enforces emulated computation throughout the analysis to assess accuracy independent of the target hardware. Low precision can be applied to model parameters, to the computations performed by the models and to the activation maps that are passed between operators. Here we follow the extrinsic quantization approach [30], where we enforce a precision caused by the reduced type $T_{w,t}$ of storage width $1 + w + t$ to be applied to all model parameters and all activation maps that are passed between operations. Our analysis follows the IEEE 754 standard [57], which defines storage encoding, special cases (Nan, Inf), and rounding behavior of floating-point data. A sign $s$, an exponent $e$ and the significand $m$ represent a number $v = (-1)^s * 2^e * m$, where the exponent field width $w$ and the trailing significant field width $t$ limit dynamic range and precision. Types $T_{5,10}$ and $T_{8,23}$ correspond to standard formats *half* and *float*. Our experiments are based on a PyTorch [1] integration of the GPU quantization kernel based on the high-performance floatx library [16], which implements the type $T_{w,t}$. A fast precision analysis allows us to evaluate more than 3,000 models with a full grid search of 184 types ($w \in [1, 8], t \in [1, 23]$) of the entire validation data.

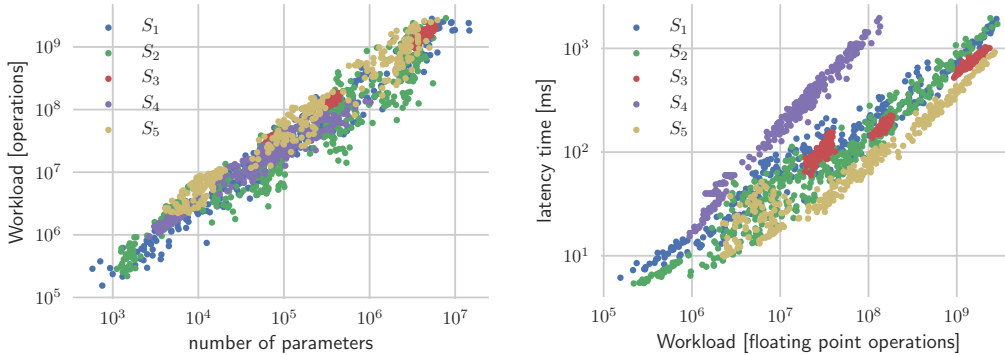

Figure 2: Left: High correlations between two analytical properties of network architectures. Right: Runtime-dependent latency is best correlated with the workload when different search space-specific characteristics are present.

## 3.3 Deployment on hardware and performance characterization

To evaluate model execution performance on the IoT target device, we perform a calibration to assess the execution speed of the models of interest. Despite many choices of deep-learning frameworks, ways of optimizing code depending on compilation or software version and even several hardware platforms that accelerate deep learning models, we formulate the performance characterization in a general manner and as decoupled as possible from the topology architecture search and the precision analysis to facilitate subsequent extensions. Performance measurements on the IoT device are affected by explicit and implicit settings. We demonstrate our search algorithm with performance measurements featuring the fewest assumptions and requirements regarding runtime. To that end, we selected Raspberry-Pi 3(B+) as a representative low-cost IoT device. It features a Broadcom BCM2837B0, quad-core ARMv8 Cortex-A53 running at 1.4 GHz. The board is equipped with 1 GB LPDDR2 memory [2]. The Raspberry-Pi 3(B+) belongs to the general-purpose device category that is shipped with peripherals (WiFi, LAN, Bluetooth, and USB, HDMI) and a full operating system (Raspbian, a Linux distribution). It is available for about $35 [32].

Throughout this work, we measure the model inference latency on the target device by averaging over ten repetitions. We used a batch size of one to minimize latency and internal memory requirements. The latency study covers many relevant use cases, for example the classification of sporadically arriving data within a short time to prolong battery lifetime or frame processing a video stream, where the classification must be completed before the next frame arrives.

For each model, we consider two analytical properties, the number of trainable parameters and the workload measured as the number of floating-point operations required for inference. The calibration step relates analytical properties with execution performance and allows us to separate runtime metrics. Figure 2 shows high correlations between the number of parameters, the workload and the measured latency on the Raspberry-Pi 3(B+) device. Workload and parameters follow a similar scaling over five orders of magnitude with homogeneous variations. The dynamic range of the latency spans more than two orders of magnitude with higher variations for larger models. However, owing to the compute-bound nature of the kernels, the workload is a better indicator of latency time than the number of parameters.

## 4 Fast cognitive design algorithms

In this section, we leverage the architecture search, the precision analysis, and the hardware calibration steps to synthesize case-specific solutions that satisfy given constraints. We address two tasks: First, the constraint search solves for the model that best satisfies given constraints. Second, the Pareto front elaboration provides insights into tradeoffs over the entire solution space. The two tasks are related. Solving the first task on a grid of constraints provides solutions to the second task, whereas filtering the latter based on the given constraints yields the former. Both tasks are solved by manually and automatically by defining the sampling law configurations on the same set of narrow-search

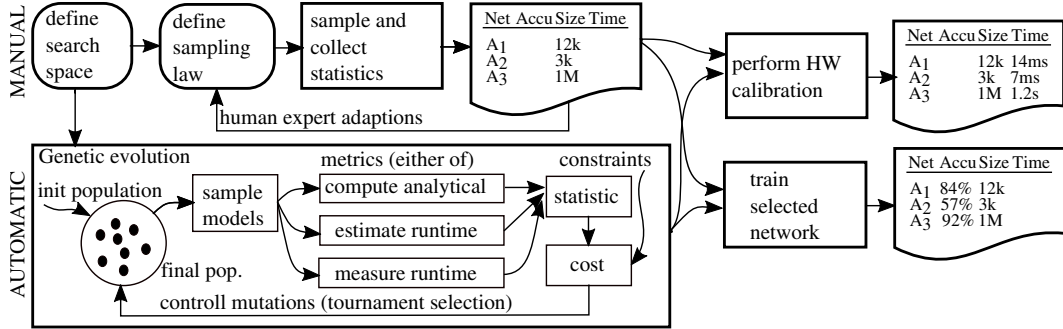

Figure 3: Manual and automatic workflow. First, sampling laws are defined to generate models of interest. Second, models are calibrated to check latency on the IoT device, even if they are not yet trained. Third, models are trained to achieve accuracy. As training is the most expensive task, it is essential to limit the number of trained models to candidates of interest only.

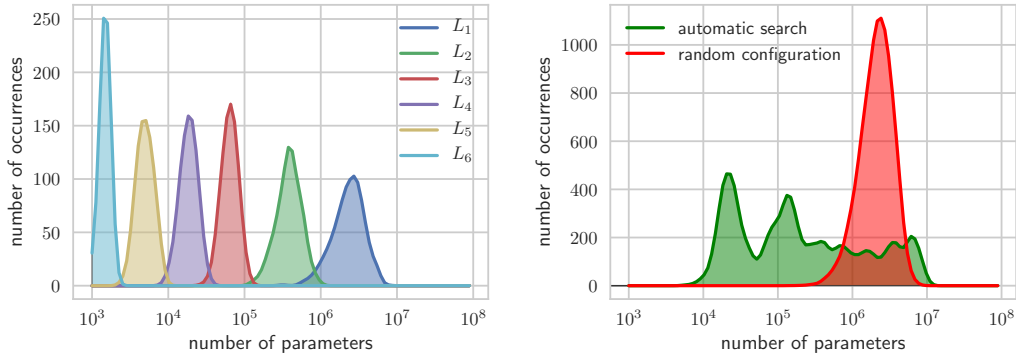

Figure 4: Left: Manually defined restricted sampling laws cover the entire space. Right: Automatic search finds sampling law configurations without human interaction and the distribution covers a higher dynamic range than sampling uniformly in the entire space.

spaces as shown in Figure 3. In the manual task, collected statistics of analytical network properties provide quick feedback to adapt the settings to cover the range of interest. For a fair comparison of the manual and automatic workflows, we assume throughout our experiments that the expert has no further feedback knowledge about model accuracy. Additionally, network runtime performance metrics can be measured on the target device or estimated from calibration measurements. Next, depending on the task type, either a few candidate networks that satisfy constraints or a full wave of networks are selected for training. Large-scale training takes the most time—as each training job is of complexity $O(n_{train}C_{model}E)$—proportional to the amount of training data, model complexity and the number of epochs for which the model is trained.

We designed a genetic and clustering-based algorithm to automatize the design of sampling laws. We define the valid space with a list of variables with absolute minimal and maximal ratings. A sampling law $L(S_i)$ is defined as an ordered set of uniform sampling laws $L = (U_x[l_x, h_x], ...)$ with lower and upper limits $l_x$ and $h_x$ per variable $x$. The genetic algorithm automatically learns the search space-specific sampling law limits $[l_x, h_x]$. The cost function is defined in a two-step approach. First, the statistic $(\mu_m, \sigma_m) := E_m^n(L)$ is estimated by computing means and standard deviations over the metric $m$ extracted from the $n$ generated topologies. Second, cost is computed as $c((\mu_m, \sigma_m), (\tau_1, \tau_2)) := |\mu_m - \sigma_m - \tau_1| + |\mu_m + \sigma_m - \tau_2|$ in order that the high density range of the estimated distribution coincides with a given interval $(\tau_1, \tau_2)$. We avoided definitions based on single-sided constraints such as $\mu < \tau$ because such formulations might be satisfied trivially (using the smallest network) or by undesirable laws having wide or narrow variations. We used the tournament selection variant of genetic algorithms [18] and defined mutations by randomly adapting the sampling law of hyper-parameters $l_x$ and $h_x$. We used an initial population of $n_{init} = 100$ and ran the algorithm

for $n_{steps} = 900$ steps while using $n_{eval} = 10$ samples to estimate mean and standard deviations per configuration. This way, one search considers $(n_{init} + n_{steps}) * n_{eval} = 10,000$ networks. As the final population might contain different sampling laws of similar quality, we performed spectral clustering [43] to find $k = 10$ clusters with similar sampling laws. We assembled a list of the most different top-$k$ laws by taking the best-fit law per cluster.

To elaborate the entire search space with a Pareto optimal front, we split each decade into three intervals $[\tau, 2\tau, 5\tau, 10\tau]$ and define a grid for $\tau = 10^3, 10^4, 10^5, 10^6$ spanning five orders of magnitude. We ran the genetic search algorithm several times by setting the target bounds $(\tau_1, \tau_2)$ in a sliding-window manner over consecutive values from the defined grid. Finally, we accumulated results from twelve genetic searches, each of which found ten sampling laws, where we sampled each law $n_{val} = 100$ times to obtain the statistic of $12,000$ network architectures per narrow-space search. Figure 4 shows results for manually and automatically sampled networks. Even though the manual search covers the region of interest nicely, human expertise is required to define the parameters of the laws $L_1$ to $L_6$ correctly. The naive sampling approach in the entire search space produces a narrow distribution and is strongly skewed towards larger networks. In contrast, the genetic algorithm equalizes the distribution and provides samples that cover much higher dynamic ranges, extending the scale especially for smaller networks without manually restricting the architecture.

## 5 Results

To study our algorithm, we ran full design-space explorations on the well-established CIFAR-10 [28] classification task and compared our results with those obtained with established reference models. Figure 5 shows the tradeoff between model size and accuracy, including manually and automatically generated results of the aggregate search spaces. The Pareto optimal front follows a

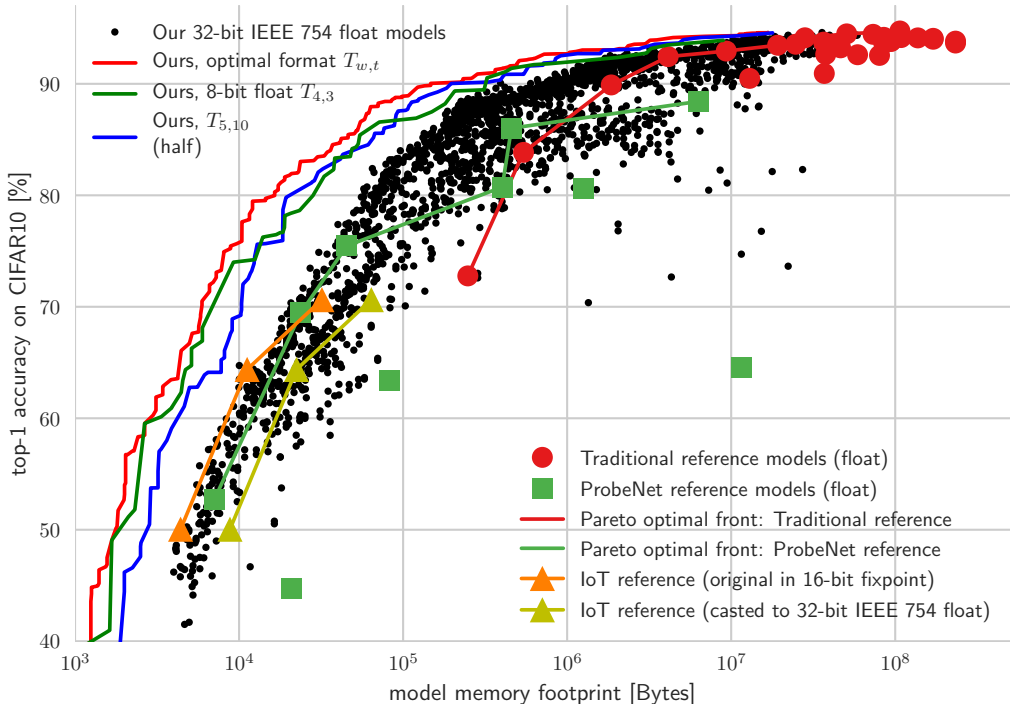

Figure 5: Results of our architecture search compared with reference models. Each dot represents a model according to its size and the obtained accuracy on the CIFAR-10 validation set. Our search finds results over five orders of magnitude and, in particular, finds various models that are much smaller than out-of-the box models. In the restricted IoT domain, our search delivers models that outperform the reference with a wide margin for fixed constraints.

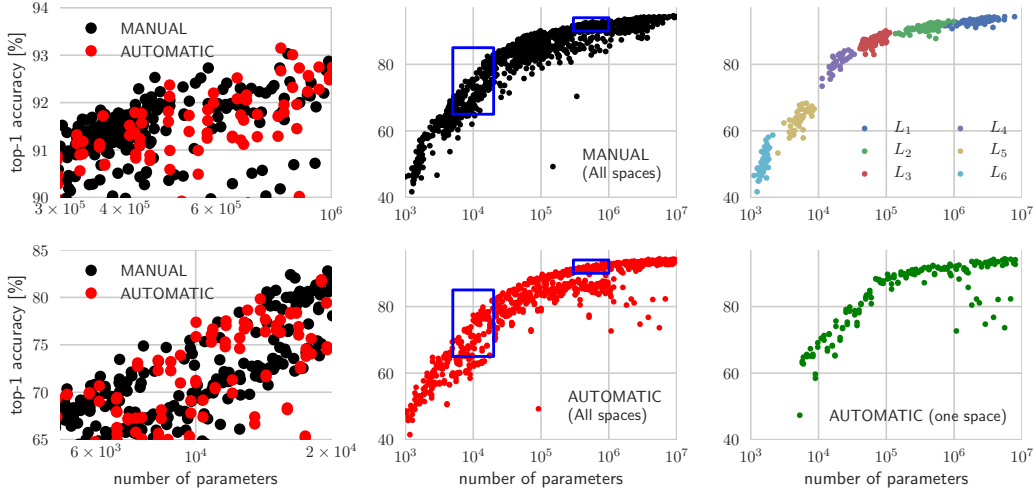

Figure 6: Left: Zoomed view of direct comparison; manual and automatic searches perform equally well. Middle: Manual and automatic search results. In the manual case, clusters are visible, whereas the automatic search sampled in a more homogeneous manner. Right: Results for one narrow-space search with marked clusters matching Figure 4.

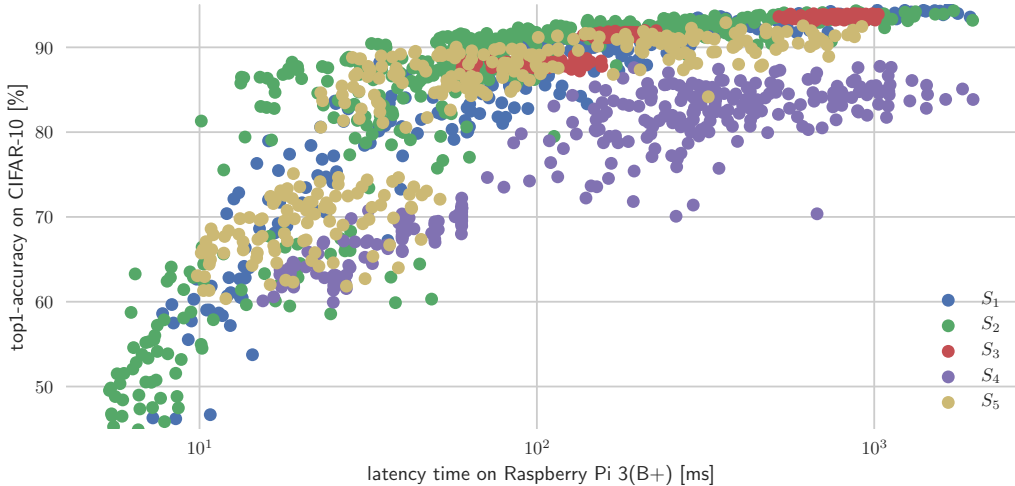

Figure 7: Final result showing the achievable tradeoffs between the IoT device measured model latency and model accuracy. Our search is able to deliver models that run below 10 ms on a Raspberry Pi 3(B+), which we took as a representative low-cost IoT device.

smooth curve that saturates towards the best accuracy obtainable for large models. The number of parameters is logarithmic and the accuracy scales linearly. Even very small models with fewer than 1000 parameters can achieve accuracies of greater than 45%. The accuracy increase per decade of added parameters is on the order of 30%, 15%, 3% and < 2% points and then decreases very quickly. This effect allows us to construct models having several orders of magnitude fewer parameters. It also provides economically interesting solutions for IoT devices that are powerful enough to process data in real time. We compare our results with three sources of reference models: (a) traditional reference models, (b) ProbeNets [40] that are designed to be small and fast and (c) models designed to run on the parallel ultra-low power (PULP) platform [11]. Traditional models include 30 reference topologies including variants of VGG [41], ResNets [20], GoogleNet [45], MobileNets [23] dual-path nets (DPNs) [8] and DenseNets [25], where most of them (28/30) exceed 1 M parameters. ProbeNets

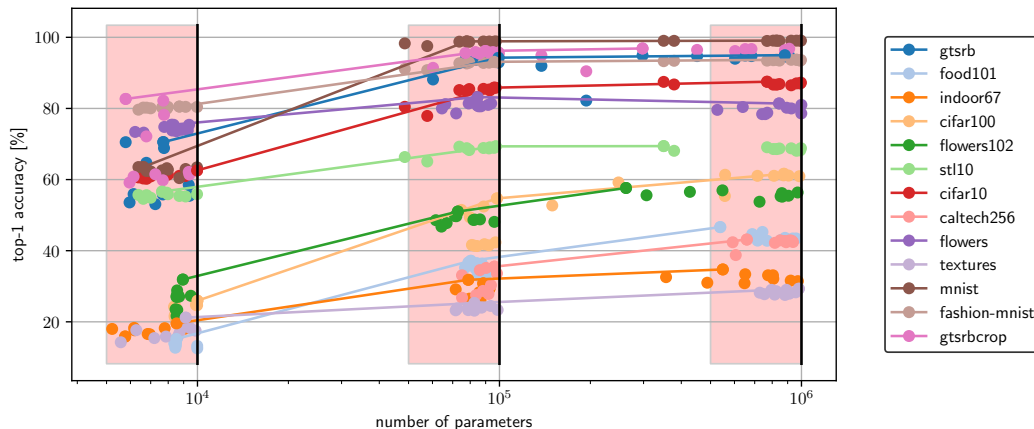

Figure 8: We demonstrate the scalability of our approach by applying our search to three constraints on thirteen datasets. Best models per dataset and constraint are connected with a line.

were originally introduced to characterize the classification difficulty and are considerably smaller by design [40]. They act as reference points for manually designed networks that cover the relevant lower tail in terms of parameters. In the IoT-relevant domain (<10 M parameters), our search outperforms all the listed reference models.

The top three fronts in Figure 5 show the results of our precision analysis. For each trained model, we evaluated the effect of running models with all configurations of type $T_{w,t}$ and plot the Pareto-optimal front. We considered three cases: (1) running all models with half-precision, (2) running all models with the type $T_{4_3}$, which is the best choice for types that are 8 bits long, and (3) running each model with its individual best tradeoff type $T_{w,t}$. We demonstrate empirically that reduced precision pushes the Pareto optimal front. Under a given memory constraint, accuracy improves by more than 7% points for half and by another 1% points or more for the model individual format.

Figure 6 shows details of manual and automatic searches, both of which yield very similar results. The right-hand graphs show results obtained for one narrow-space search, where manually defined sampling laws led to clusters. The automatic search covered a similar range homogeneously. Figure 7 shows inference times when the same set of models is executed on a Raspberry Pi 3(B+). Similarly, providing additional latency time for small models results in dominant accuracy gains, however, large models only slightly improve accuracy even when using more complex models that require long evaluation times.

Figure 8 demonstrates the scalability of our approach. We applied our search for three constraints $\tau = 10^3, 10^4, 10^5$ on thirteen datasets [40], where we spent a training effort of ten architectures per dataset and constraint. The lines connect the best per constraint and dataset performing architectures.

## 6    Conclusion

We studied the synthesis of deep neural networks that are eligible candidates to run efficiently on IoT devices. We propose a narrow-space search approach that leverages knowledge quickly from existing architectures and that is modular enough to be adapted to new design patterns. Manually and automatically designed sampling laws allow various models to be generated having sufficiently numerous parameters to cover multiple orders of magnitude. We demonstrate that reduced precision improves top-1 accuracy by over 8% points for constraint weight memory in the IoT-relevant domain. A strong correlation between model size and latency enables us to create small enough models that provide superior inference response latencies below 10 ms on an edge device costing only about $35.

**Acknowledgments**

This work was funded by the European Union's H2020 research and innovation program under grant agreement No 732631, project OPRECOMP.

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
