[Supplementary Material]

# Appendix for:
# 'Constrained deep neural network architecture search for IoT devices accounting hardware calibration'

1  The following Sections cover technical details of our work. Section 1 defines the search spaces
2  and manually designed sampling laws, Section 2 provides statistics over networks sampled from
3  the defined laws, Section 3 details data augmentation and hyper-parameters used for training and
4  Section 4 justifies the choice of the IoT device and explains the workflow to export a model from
5  the back-end framework and to deploy it on the IoT device. Section 5 elaborates how we scale our
6  algorithm to multiple datasets and provides further information of the datasets.

## 1  Search space and sampling law definition

Table 1: Search spaces induced from established reference models

| Space | Reference model | Params | Ops | Our Acc[1] | Ref Acc[2] [5] |
|-------|-----------------|--------|-----|---------|-------------|
| $S_1$ | DenseNet121 [13] | 7.0M | 898.1M | 94.13% | 95.04% |
| $S_2$ | MobileNetV2 [12] | 2.3M | 94.6M | 92.94% | 94.43% |
| $S_3$ | GoogLeNet [20] | 6.2M | 1.5G | 93.55% | - |
| $S_4$ | PNASNetA [15] | 135.5K | 29.2M | 83.85% | - |
| $S_5$ | ResNeXt29_32x4d [22] | 4.8M | 779.6M | 93.46% | 94.73% |

[1] reproduced results with our training limited to 100 epochs
[2] reference results of third-party implementation with high-effort training of 350 epochs

8  Table 1 lists the five search spaces used in this work that are based on established models. Typical
9  models consist of 2M up to 7M of parameters and cause workloads from 94.6 million up to 1.5 billion
10 FLOPs and are too large for fast implementations on a targeted IoT device. DenseNets [13] exists in
11 common variants, 121, 161, 169, and 201 and we used the smallest variant (DenseNet121) as starting
12 point. We reproduced the accuracy for all architectures by running our training procedure as detailed
13 in Section 3 where we used an upper limit of 100 epochs and compare it with the claimed reference
14 accuracy from the source from where we obtained the architecture implementation in PyTorch [3].
15 The latter values are slightly higher but they are obtained with a high effort training that runs for a
16 fixed amount of 350 epochs. Additionally, the later source does not state the mean and variance of
17 the training process neither is it completely clear if the values are obtained in a one-shot training or if
18 the best values have been selected after repeating the training process several times. In contrast, we
19 decided to follow a pragmatic but efficient approach of evaluating each architecture only once and
20 to limit training effort to an affordable value of 100 epochs. This decision is motivated by the fact
21 that we want the same training procedure to be applied to over 3'000 models. Training evaluations
22 with high-effort would cause $3.5\times$ more computational costs and repeating experiments to deliver
23 statistics would at least require a repetition factor of $5\times$. Both aspects together cause a $17.5\times$ increase
24 in computation cost. In our opinion, if we are willing to pay such an increase, it would be more
25 interesting to use an affordable approach and invest the additional budget into investigating more

26  architectures. The increased effort would allow investigating over 50'000 network architectures. Next,
27  we define the sampling laws and parameters used to manually enforce smaller variants of networks
28  within the defined spaces.

29  DenseNets [13] consists of four stages, each repeating DenseNet unique blocks. We identified the
30  stage-specific number of repetitions, the growth rate and the reduction factor as relevant hyper-
31  parameters that we modify. Table 2 specifies the sampling laws. In this case, we decided to clip the
32  repetition factor at 32 which additionally includes the configuration of DenseNet161. The normalized
33  reduction factor is sampled with a step size of 0.01.

Table 2: Architecture search space definition $S_1$ with different sampling laws for DenseNets

| Law | Cardinality | Densenet parameters | | |
|---|---|---|---|---|
| | | $n_i, i \in \{1,2,3,4\}$ | $g^*$ | $r^{**}$ |
| $L_0$ | $3.3 * 10^9$ | $[1,32]$ | $[1-32]$ | $[0.0,1.0]$ |
| $L_1$ | $2.0 * 10^6$ | $[1,8]$ | $[1-8]$ | $[0.2,0.8]$ |
| $L_2$ | $6.3 * 10^7$ | $[1,16]$ | $[1-16]$ | $[0.2,0.8]$ |
| $L_3$ | $1.0 * 10^9$ | $[1,32]$ | $[1-32]$ | $[0.5,0.8]$ |

$^*$ $g$ is the growth rate,    $^{**}$ $r$ is the reduction rate

34  MobileNetsV2 [12] consists of seven stages, each repeating MobileNetsV2 unique blocks. Each block
35  is configured with four parameters, input number of channels, output number of channels, expansion
36  factor, and a stride factor. To generate valid configurations, we define the first input channel number
37  separately, since the subsequent input shape follows directly from the previous block configuration.
38  Additionally, we restrict the stride factor per stage to either be one or two and we further require to
39  sample exactly three twos and four ones. The reason for this choice is due to the stride parameter
40  directly influences the spatial shape of tensors and the limitation ensures fixed downsampling over
41  three steps from $32 \times 32$ to $4 \times 4$. Additionally, the existing intermediate last convolutional layer is
42  separately parametrized and is used as in the original reference model as a transition layer between
43  the last block and the final linear classifier. Table 3 states the sampling law definitions.

Table 3: Architecture search space definition $S_2$ with different sampling laws for MobileNets

| Law | Cardinality | MobileNet parameters, $i \in [1,7]$ | | | | | |
|---|---|---|---|---|---|---|---|
| | | $f_{in}$ | $e_i$ | $f_i$ | $n_i$ | $s_i$ | $f_{out}$ |
| $L_0$ | $5.2 * 10^{34}$ | $[1,128]$ | $[1,8]$ | $[1,256]$ | $[1,4]$ | $[1,2]$ | $[1,1280]$ |
| $L_1$ | $3.6 * 10^{33}$ | $[16,128]$ | $[1,6]$ | $[16,256]$ | $[1,4]$ | $[1,2]$ | $[128,1280]$ |
| $L_2$ | $2.0 * 10^{28}$ | $[16,64]$ | $[1,4]$ | $[16,128]$ | $[1,3]$ | $[1,2]$ | $[128,512]$ |
| $L_3$ | $3.1 * 10^{21}$ | $[16,32]$ | $[1,2]$ | $[16,64]$ | $[1,2]$ | $[1,2]$ | $[128,256]$ |
| $L_4$ | $4.0 * 10^{19}$ | $[16,32]$ | $[1,2]$ | $[4,32]$ | $[1,2]$ | $[1,2]$ | $[64,128]$ |
| $L_5$ | $1.4 * 10^{15}$ | $[16,32]$ | $[1,2]$ | $[2,8]$ | $[1,2]$ | $[1,2]$ | $[16,64]$ |
| $L_6$ | $4.3 * 10^{13}$ | $[4,8]$ | $[1,2]$ | $[2,8]$ | $[1,2]$ | $[1,2]$ | $[12,16]$ |

44  GoogLeNet [20] is composed of the characteristic Inception module, which is defined through seven
45  intermediate channel depths. The full network is grouped into three stages, first a convolutional
46  pre-layer, second, and third a max-pooling separates sequences that are built from two, five, and two
47  Inception modules. Table 4 defines the sampling laws. We choose parameter specific upper bounds
48  oriented on the reference implementation.

49  PNASNet-A [15] consists of three stages that are build by repeating cell-A type of blocks. The stages
50  are separated by downsampling layers that are implemented as cell instances with a stride of two.
51  Table 6 defines the sampling laws that affect the number of block repetitions and the number of
52  channels used in the block, where $f_2$ and $f_3$ are relatively defined to the output shape of previous
53  stages.

Table 4: Architecture search space definition $S_3$ with different sampling laws for GoogLeNetd

| Law | Cardinality | $f_0^0$ | $f_i^1$ | $f_i^2$ | $f_i^3$ | $f_i^4$ | $f_i^5$ | $f_i^6$ |
|---|---|---|---|---|---|---|---|---|
| $L_0$ | $3.4*10^{122}$ | [16,256] | [16,384] | [16,192] | [16,384] | [16,48] | [16,128] | [16,128] |
| $L_1$ | $2.8*10^{119}$ | [16,256] | [16,384] | [16,192] | [16,384] | [16,48] | [16,128] | [16,128] |
| $L_2$ | $1.1*10^{97}$ | [16,256] | [16,64] | [16,64] | [16,64] | [16,32] | [16,32] | [16,32] |
| $L_3$ | $9.8*10^{51}$ | [16,256] | [16,32] | [8,16] | [16,32] | [4,8] | [4,8] | [4,8] |
| $L_4$ | $6.3*10^{39}$ | [16,128] | [4,8] | [4,8] | [4,8] | [4,8] | [4,8] | [4,8] |
| $L_5$ | $5.2*10^{26}$ | [8,16] | [4,6] | [4,6] | [4,6] | [4,6] | [4,6] | [4,6] |
| $L_6$ | $5.0*10^{38}$ | [8,16] | [2,6] | [2,6] | [2,6] | [2,6] | [2,6] | [2,6] |

$f_{i+1}^0 := f_i^1 + f_i^3 + f_i^5 + f_i^6$ for $i \geq 0$, recursive definition such that next input shape matches the previous output shape

Table 5: Architecture search space definition $S_4$ with different sampling laws for PNASNet-A

| Law | Cardinality | $n_i$ | $f_1$ | $d_2$ | $d_3$ |
|---|---|---|---|---|---|
| $L_0$ | $3.5*10^6$ | [1,12] | [1,128] | [1,4] | [1,4] |
| $L_1$ | $3.1*10^6$ | [1,12] | [16,128] | [1,4] | [1,4] |
| $L_2$ | $2.3*10^5$ | [1,8] | [16,64] | [1,3] | [1,3] |
| $L_3$ | $9.7*10^2$ | [1,3] | [8,16] | [1,2] | [1,2] |

Set $f_2 := f_1 * d_2$ and $f_3 := f_2 * d_3$.

ResNeXt [22] is the improvement over the typical ResNet [11] structure. It consists of a three-stage architecture where each stage repeats the bottleneck block $n_i$ times, for $i \in \{1,2,3\}$. The block consists of the typical residual connection and follows a bottleneck design where grouped convolutions are used to reduce the kernel size. We define in the search space with the bottleneck base width $f_i$ and the cardinality $c_i$. However, the total channel size that is invoked during the grouped convolution operation is of width $f_i * c_i$. Since we want to limit the product $f_i * c_i$ but we also require it to be devisable by either $f_i$ and $c_i$ we decided to randomly sample the later and restrict the cardinality upper bound to be $l_{high}/f_i$, where $l_{high}$ denotes the upper product limit and $f_i$ is dependent on the current sampling of the base depth. Table 6 summaries the defined random laws.

Table 6: Architecture search space definition $S_5$ with different sampling laws for ResNeXt

| Law | Cardinality | $n_i$ | $f_i$ | $c_i$ |
|---|---|---|---|---|
| $L_0$ | $9.5*10^{14}$ | [1,3] | [1,64] | [1,512] |
| $L_1$ | $2.1*10^{10}$ | [1,3] | [4,64] | [1,512/$f_i$] |
| $L_2$ | $2.4*10^8$ | [1,3] | [4,32] | [1,128/$f_i$] |
| $L_3$ | $1.5*10^5$ | [1,2] | [4,8] | [1,32/$f_i$] |

## 2 Statistical results of used networks

For the defined search spaces and sampling laws, we collected statistics over 1000 networks that are presented in Figure 1. We targeted to cover the full domain in $[10^3, 10^7]$. Some search spaces, such as $S_1$, $S_3$, and $S_5$ are quickly covered with three simple configurations. Other search spaces, such as $S_2$ and $S_3$, lead to narrower distributions. We decided to add three additional sampling laws to cover the lower domain. The right-hand side of Figure 1 shows statistics obtained when networks

(a) Manual laws on $S_1$

(b) Automatic law on $S_1$

(c) Manual laws on $S_2$

(d) Automatic law on $S_2$

(e) Manual laws on $S_3$

(f) Automatic law on $S_3$

are obtained by uniformly sampling each parameter in its full domain (according to the law $L_0$ in the previous tables) and when they are obtained with automatically generated sampling laws that are adjusted with our proposed genetic algorithm. The base law of the original definition has a high impact on where the actual mass of the distribution concentrates. The main densities are around $10^6$ parameters for $S_1$ and $S_2$. $S_3$ and $S_4$ have the center of mass above $10^7$ parameters which cause difficulties for the genetic algorithm to converge towards the low end of the domain. Still, in all cases, the genetic algorithm is capable of either considerably moves the center of mass to the left or to flatten out the distribution. For completeness, Table 7 states the parameter and flop metrics obtained with our defined search spaces.

(g) Manual laws on $S_4$

(h) Automatic law on $S_4$

(i) Manual laws on $S_5$

(j) Automatic law on $S_5$

Figure 1: Statistic over manual (left) and automatic generated (right) networks for all search spaces $S_1$ up to $S_5$. With manually design the sampling law, a human expert can reasonably adjust and focus the distribution into regions of interest, either close to a target constraint and in a general way to cover five orders of magnitude.

## 3 Training setup

We conducted all training experiments in a controlled environment where we trained from scratch for each candidate architecture. We used PyTorch version 0.4.1 as development framework and run on IBM Power8 or Power9 nodes equipped with either P100 or V100 GPUs. We used standard on-the-fly data augmentation during training that pads images with 4 pixels and randomly crops the image to $32 \times 32$ pixels, apply horizontal flipping with a probability of 0.5 and finally normalizes pixel values to zero mean and unit variance. During testing, the original $32 \times 32$ images are directly normalized and feed into the models. For training, we used stochastic gradient descent with a batch size of 128 samples configured with an initial learning rate of 0.01, a momentum of 0.9, and a weight decay factor of $5 * 10^{-4}$. We used a fixed scheduling schema where the learning rate is divided by a factor of 10 at epoch 40 and 70 and we limit training to stop at 100 epochs.

## 4 Deployment setup

In our work, we decided to demonstrate our algorithm to produce optimized network architectures for the Raspberry-Pi 3(B+) as stated in our paper. At this point, we consider it worth to justify our choice. First, it should be mentioned that there is a current emerging trend in industry and research that pushes to improve hardware for artificial intelligence (AI) by either improving performance, reducing power consumption or providing better trade-offs in terms of power/performance ratios or hardware cost versus the on-device supported features. In terms of thinking through IoT driven business cases, the fact that new hardware appears requires to benchmark and rethink on what HW

Table 7: Overview of parameters and flop metrics of generated architecture search spaces. We started from seven well-known topologies and defined at least three sampling laws per narrow search. The generated search spaces cover over three order of magnitudes in the amount of weights and flops.

| Search space | | Number of parameters | | | Number of flops | | |
|---|---|---|---|---|---|---|---|
| | | Min | Mean+/-Std | Max | Min | Mean+/-Std | Max |
| DenseNet | L1 | 5.8e+2 | 2.6e+4+/-2.3e+4 | 1.2e+5 | 1.5e+5 | 7.5e+6+/-7.2e+6 | 3.3e+7 |
| | L2 | 0.e+0 | 2.3e+5+/-2.3e+5 | 1.6e+6 | 0.e+0 | 6.9e+7+/-7.6e+7 | 4.0e+8 |
| | L3 | 5.6e+3 | 2.8e+6+/-3.e+6 | 1.9e+7 | 4.5e+5 | 7.6e+8+/-9.1e+8 | 4.8e+9 |
| MobileNetV2 | L1 | 9.5e+2 | 1.6e+3+/-2.0e+2 | 2.2e+3 | 1.8e+5 | 4.5e+5+/-1.7e+5 | 1.2e+6 |
| | L2 | 2.5e+3 | 5.4e+3+/-1.4e+3 | 9.6e+3 | 9.e+5 | 2.8e+6+/-1.2e+6 | 6.6e+6 |
| | L3 | 8.8e+3 | 2.0e+4+/-5.1e+3 | 4.0e+4 | 1.4e+6 | 5.9e+6+/-3.2e+6 | 2.1e+7 |
| | L4 | 2.8e+4 | 6.9e+4+/-1.6e+4 | 1.3e+5 | 2.2e+6 | 1.5e+7+/-1.1e+7 | 7.2e+7 |
| | L5 | 1.2e+5 | 4.3e+5+/-1.5e+5 | 9.9e+5 | 8.6e+6 | 1.0e+8+/-8.4e+7 | 6.9e+8 |
| | L6 | 3.1e+5 | 2.7e+6+/-1.2e+6 | 7.8e+6 | 2.1e+7 | 6.3e+8+/-5.7e+8 | 4.0e+9 |
| GoogLeNet | L1 | 4.5e+3 | 6.5e+3+/-5.4e+2 | 8.3e+3 | 1.8e+6 | 2.6e+6+/-2.8e+5 | 3.4e+6 |
| | L2 | 8.8e+3 | 9.8e+3+/-3.2e+2 | 1.1e+4 | 3.3e+6 | 3.8e+6+/-1.9e+5 | 4.3e+6 |
| | L3 | 1.3e+4 | 1.7e+4+/-2.e+3 | 2.2e+4 | 4.9e+6 | 8.7e+6+/-1.9e+6 | 1.3e+7 |
| | L4 | 4.5e+4 | 6.4e+4+/-6.3e+3 | 8.0e+4 | 1.6e+7 | 2.9e+7+/-5.7e+6 | 4.3e+7 |
| | L5 | 2.8e+5 | 3.8e+5+/-3.2e+4 | 4.9e+5 | 1.0e+8 | 1.5e+8+/-1.9e+7 | 2.1e+8 |
| | L6 | 2.4e+6 | 4.2e+6+/-6.6e+5 | 6.8e+6 | 8.0e+8 | 1.5e+9+/-3.1e+8 | 2.5e+9 |
| PNASNet | L1 | 2.7e+3 | 4.8e+3+/-1.2e+3 | 7.6e+3 | 9.4e+5 | 1.8e+6+/-4.9e+5 | 2.8e+6 |
| | L2 | 6.8e+3 | 6.0e+4+/-3.6e+4 | 2.1e+5 | 2.4e+6 | 1.6e+7+/-8.e+6 | 4.4e+7 |
| | L3 | 1.6e+4 | 2.6e+5+/-2.2e+5 | 1.5e+6 | 4.1e+6 | 5.1e+7+/-3.3e+7 | 2.e+8 |
| ResNeXt | L1 | 3.0e+3 | 1.1e+4+/-3.9e+3 | 2.2e+4 | 1.6e+6 | 6.1e+6+/-2.6e+6 | 1.2e+7 |
| | L2 | 1.5e+4 | 1.6e+5+/-7.4e+4 | 4.5e+5 | 3.7e+6 | 7.e+7+/-4.e+7 | 2.1e+8 |
| | L3 | 1.0e+5 | 2.1e+6+/-1.1e+6 | 6.8e+6 | 2.1e+7 | 8.5e+8+/-5.8e+8 | 2.7e+9 |

product a certain IoT application should be built. We are aware of the existence of tens of ASIC or FPGA solutions that might be selected for business legitimated reasons as a target edge inference system.

We think that the crucial factors for a successful IoT deployment strategy cover the following points with an importance that is application specific:

- reliability
- user/developer friendly software ecosystem
- modular integration or extensions of different functionality
- typical IoT support
- cost efficient system

We decided that in this work we focus on the algorithm. However, since we are aware of many choices and good reasons for a certain HW solution, we developed our approach such that the main functionality is decoupled from the actual HW implementation. Especially, populating a database with the current results that are obtained with expensive training for obtaining the model accuracy, can easily be reused later on for any hardware platform by just implementing the inference and timing measurement setup in order to obtain the new HW calibration information. In this work, we focus on the most general use-case that causes the least amount of requirements for the underlying hardware. That way, we identified the Raspberry-Pi 3(B+) as general purpose quad-core architecture as a suitable IoT device candidate. The Raspberry-Pi proves its marketability by the fact that it has been shipped over 25 million times by February 2019 [4]. Even though there are competing products that are specially tailored for AI deployment, the choice of selecting a general-purpose platform equipped with a Linux operating system comes with obvious advantages, such that it enables to reuse established software and solutions can be easily extended to any needs. In contrast to dedicated AI accelerators that are shipped as USB dongles, potentially required features such as Ethernet, WiFi, SDCard slot, or USB ports are already included in the Raspberry-Pi 3(B+). Even though we are aware that a general purpose architecture cannot compete in some performance metrics with a dedicated AI

product, we argue that our work is especially insightful since we cover the more challenging case on optimizing for a performance limited device. In our view, it is plausible enough to argue that a more performant device will automatically deliver better results. We aim to support various HW platforms with different deployment flows in future work.

Next, we describe the deployment flow for the Raspberry-Pi 3(B+). Even though our back-end algorithms, as well as our training routine, is implemented in PyTorch, we still aim to remove the back-end dependency in order to be open and to ease later migration to new target platforms, frameworks, and ecosystems. To that end, we decided to export all models according to the open neural network exchange (ONNX) format [2]. We decided to use caffe2 as target device runtime for the exported ONNX models. We build the caffe2 framework directly from a full source compilation with all default parameters on the Raspberry-Pi 3(B+) and we ensured that the produced code is using the ARMs NEON library [1] for fast computation. We wrote a light script to import the produce ONNX models and we trigger a sequence of inferences for a single image. In our work, all timing results have been obtained by averaging wall clock times over ten runs.

## 5  Datasets

Our large-scale search that provides Pareto optimal fronts is conducted on CIFAR10 [14] with the provided train and test splitting. We demonstrate the scalability of our architecture search by applying a fast customized search to individual datasets. In contrast to previous architecture searches that include training inside the main optimization loop, we can sample a very large amount of neural networks in a short time *without* training. Additionally, we can run the genetic algorithm to bias the sampling process into domains we are interested *without* any single network training step. With our approach, customized searches become affordable. For each dataset we performed the following workflow: we define an upper constraint $\tau$, we run a genetic search with the optimization goal to deliver a sampling law with a probability density function that is concentrated in $\tau_1 = 0.5\tau, \tau_2 = \tau$, we sample 100 candidate networks from the found sampling law, we filter out the good models that strictly satisfy the one-sided constraint $< \tau$ and we randomly select 10 suitable networks. Finally, only 10 candidate networks are entering the compute-intensive trained procedure. This approach allows affording to validate our algorithm on thirteen datasets for three considered constraints. Figure 8 of our paper presents the results. The following explains the considered datasets.

We focus on sixteen public available and established image classification datasets: *MNIST* [9], *GTSRB* [19], *svhn* [16], *CIFAR10* [14], *flowers*[1], *flowers102* [17], *fashion MNIST* [21], *food101* [6], *CIFAR100* [14], *stl10* [8], *textures* [7], *indoor67* [18], *caltech256* [10], *quickdraw*[2], and *places* [23]. Figure 2 shows the number of classes, Figure 3 shows the balance of the classes as ratio of samples of the majority over the minority class, and Figure 4 shows the number of samples used for training and testing. The datasets span two order of magnitudes in the number of classes and in the number of available training samples and one order of magnitude in the balance ratio. The datasets stem from various domains and cover typical and relevant use cases such as optical digit recognition stemming from handwritten samples (*MNIST*) or in the context of images stemming from house numbers (*svhn*). *GTSRB* covers traffic sign recognition, a use case that occurs in autonomous driving systems. Scene recognition aims to classify the location of where the picture was taken as whole (*indoor67* and *places*), whereas traditional classification tasks are posed around identifying a class based on a particular object present within the image. In order to limit the workload, we run our proposed algorithm on 13 out of 16 datasets that have less than 100'000 images in the training set. Results are presented in Figure 8 of our paper.

Figure 2: The number of classes per dataset cover two orders of magnitude, from as few as 5 classes up to 345 classes.

Figure 3: Ratio of class samples of majority over minority class in the training data. The balance ratio spans from 1.0 (for equally balanced datasets) up to a factor of $11.9\times$.

Figure 4: The number of samples within a given dataset used for training and testing sorted by training samples. Train and test sets are always disjoint and the splitting is given as suggested by the reference. The number of training samples spans more than two order of magnitude.

## Footnotes

[1] Available at `http://download.tensorflow.org/example_images/flower_photos.tgz`

[2] Available at `https://github.com/googlecreativelab/quickdraw-dataset`