[Reviews · NeurIPS 2019]

Reviewer 1



This paper provides an interesting new method for neural network architecture search by encoding knowledge about constraints in IoT devices. As such the authors are able to search many more architectures than previous works because they do not need to train as many networks to find the best ones for the IoT use case, and they are able to find networks which provide higher accuracy under the same latency / memory constraints as compared to previous work. This paper contains several typographical errors (35$ instead of $35), and some other errors, for example their definition of "half" and "float" in Section 3.2 is backwards. These errors are minor and do not significantly detract from the paper's content. There are also some small issues with clarity. The last paragraph in Section 3.1 is one sentence, which makes it hard to read. Also, what exactly is the relationship between the manual and automatic workflows? How do they interact with each other? And within the automatic workflow, what is the difference between the automatic search and "random configuration" as shown in Figure 4? This figure in particular was hard to understand. Overall, the paper is an interesting contribution, however I am unsure of the significance of the contribution, as it does not seem like that much of an improvement over the cited previous works.

Reviewer 2



**Originality**: The novelty of the work is that the authors search in the context of IoT devices, in terms of evaluating measured latency, memory footprint, and prediction accuracy at the same time to satisfy certain constraints. This is new compared to prior works, according to the Related Work section and my understanding. Also, their method of evolution of a parameterized search space seems also new in the context of Neural Architecture Search. **Quality**: The submission is complete and not a work-in-progress. The experiment design is reasonable, the baseline comparisons against previous methods are fair and the results support their claims. The authors did not mention the weakness of this method, and there are a number of typos yet to be fixed. **Clarity**: The method description and clear and easy to understand, since the method is not very complicated. Most of the figures are self-explainable, except some require a deeper look into the text. **Significance**: Their results show that they can search for better model hyper-parameters (kernel size, channel size) for IoT devices. The results have shown that their method is better compared to prior work. They can also deal with the target space constraint (such as memory size) by evolving the input search space, and this is useful when deployed models have such constraints. Their search for different floating point configuration also prompts hardware designers and researchers to design specialized architecture for smaller deep networks that run on CPUs. Since they target IoT devices, the results can also be useful for non-researchers with affordable hardware.

Reviewer 3



This paper studies the NAS problem in the IoT platform scenarios with specific hardware considerations. The idea is to break down the original whole search space into a set of subspaces through sampling, and the final, actual search is conducted in the typically much narrower space as the union of these sampled subspaces. In the IoT platform scenarios, specific hardware considerations are applied in sampling the space. The whole work of the paper is reported through a case study on image classification using CIFAR-10 dataset, and the better ever results are reported. The only methodological contribution as I see is the idea of breaking down the original space into subspaces and conducting the search in the narrow space as the union of the subspaces. While this could be claimed as a contribution as was done in the paper, I don’t feel comfortable with buying this idea, as I see that this methodology has a lot of open issues. For example, how many sampled subspaces are reasonably sufficient to deliver a good solution? What sort of theoretic assurance of reaching to a good or better solution using this approach, even if we force each sampled subspace to cover a reference model as was done in the paper? In terms of empirical analysis, the paper only gave one case study on image classification with CIFAR-10 dataset. In real-world IoT applications, the problems we face may way more challenging than image classification in CIFAR-10. Consequently, I am not sure how much societal impact this work could generate. Finally, I would like to comment on the presentation of the paper. The paper has a lot of grammatical errors and awkward sentences. I encourage the authors to have a serious proof-reading before sending the paper out for review. The above was my initial review. After I read the authors’ response, I am willing to buy the arguments regarding the significance and novelty of the work. Specifically, regarding the claimed improvements over SOTA, my ranking of the significance goes in the order of 1 (most, though I still have a technical reservation stated above and I am not completely convinced by the authors’ response as I was expecting that the authors would offer an insightful discussion on determining the number of subspaces as well as its relationship with the give prior such as the reference models), 3 (moderate), and 2 (least). As a result, I am raising the overall score to 6.

[Author Response · NeurIPS 2019]

**All reviewers:** *Improvement over SOTA:* Our work 1) improves results by the union of narrow-space search, 2) accounts for constraints, and 3) improves accuracy for the given constraints. The reviewers agree with the novelty of point 1. Even though reference work solves points 2 and 3 and we reuse a variant of a genetic algorithm inside the inner optimization loop of our automatic workflow (L163), we claim the uniqueness of our overall approach and results. More precisely, we provide a low effort solution to a NAS problem by decomposing it into three steps that can be solved 1) analytically, 2) by calibration measurement, and 3) by training models. We stress run times inequalities; 1 analytical evaluation (100ths per sec) $<<$ 1 calibration measurement (1sec – 100sec) $<<$ 1 model training ($>$1h even on P100 GPUs). Solving the NAS for one specific constraint, the left parts depicted in Fig. 3 can be evaluated efficiently (especially, the genetic algorithm does not involve model training). We assess 10'000 of models before training, where only a small subset of those is required to be trained.

*Novelty:* First, we extend the search range to small models (we cover five orders of magnitudes in the number of parameters) whereas reference work reports around a reference point or on a limited dynamic range. Second, we evaluate quantized models on the front of optimal models. Even though quantization and reduced precision models are established, key reference results are obtained on selected operation points. In contrast, our evaluation reveals the effects of trade-offs between quantization and NAS results for smaller networks.

*Significance:* In our opinion, the significance and innovation should not be solely judged on the subparts of contributions on their own but rather by the overall impact of the approach. Our method is simple and clean. It has been designed for practical industrial applications, targetting industrial experts with limited AI background and/or time to invest in AI expertise. We provide a flexible and economical design strategy that helps to reduce development time to design AI models for edge devices and impact industrial solutions.

**Reviewer 1:** The manual and automatic workflow are independent approaches to solve the same NAS problem and they do not interact. Fig. 4 (right), is the analog plot of Fig. 1 (right), where the "random configuration" shows the statistics of networks that are sampled in the full space, the "automatic search" are the results of running and concatenating the genetic algorithm as described in L181+. We will better clarify this in the revised version.

The closest related work is MnasNet [46] that follows a traditional NAS approach, where the feedback loop includes the expensive training step reporting a run time of 4.5 days on 64 TPUv2 devices. That is equivalent to about 640 GPUh/TPUv2 pod*24h/d*4.5d = 69'120 GPU hours. In comparison, our approach (100 selected models) leads to about 200 GPU hours (similar model size as ref.). Accounting for the explainable difference of the dataset 24x (1.2M/50k samples) but also for the training 1/20x (they: 5 epochs, we: 100 epochs) our NAS approach remains about two orders of magnitude more resource-efficient! 2) Results: all MnasNet models exceed 3.9M parameters and their latency domain spans one order of magnitude between 20ms and 140ms (MnasNet, Fig. 5). In contrast, we include the 1k up to 1M parameters range that is essential to reduce the per IoT device cost and we cover two orders of magnitude on a much weaker device in terms of latency time.

**Reviewer 2:** We agree, manual and automatic comparisons are challenging to compare due to the expert knowledge. In our results, we assume a best-effort approach with an experienced data scientist. All decisions taken by the expert are listed in the appendix. We stress, that for a fair comparison, the narrow-search space is common for both approaches. Knowing Fig. 6 implies you have already spent the experimenting runtime, so if your intention is to automate and shorten the overall development time you aim to not depend on the "human in the loop" at that stage. If you still do, most probably you want to extend the search space to account for new design patterns, if you do so, for a fair comparison, you should rerun both workflows on the new space. If your initial assumption was right, and you indeed have a stronger search space, we expect to improve results in both cases.

**Reviewer 3:** The number of sampled narrow spaces that are needed to deliver a good solution depends directly on the quality of the search space. If it would be a priori clear, or there is strong evidence that one space is considerably better than others, it would be enough to perform all considerations on that space only. However, since we do not have such evidence prior to run the experiments, we propose the aggregation of multiple subspaces. Since each definition involves paper and pen definitions and related implementations the number of considered narrow spaces must be reasonably small. We conducted all experiments with five narrow-spaces.

We do not provide theoretical guarantees. In contrast, we adopt the in-the-field practice of empirically reporting results. Our opinion is that the main contribution of our work is formulating the NAS problem in a decomposed way that allows constraint evaluations and HW measurements to rule out candidates before they are required to be trained.

We conducted the key results on CIFAR-10 since we know that this is a well-established dataset with many reference results, especially experiments conducted in IoT settings on non-standard hardware. We hope that this choice allows a common ground between HW developers, the deep learning community, and the IoT industry to better judge our achievements. Fig. 8 of our paper includes results where we applied our search strategy to additional thirteen image classification tasks, as detailed in the appendix.

[Meta-Review · NeurIPS 2019]

The authors propose a novel approach to architecture search that explicitly takes computational hardware constraints into account. The writing is clear in spite of typos, the method simple, and the results convincing.